SciPost Physics
Submission

# Enstrophy without boost symmetry

N. Pinzani-Fokeeva[1,2*], A. Yarom[3]

**1** Center for Theoretical Physics, Massachusetts Institute of Technology, Cambridge, MA 02139, USA
**2** Department of Physics and Astronomy, University of Florence, Via G. Sansone 1, I-50019, Sesto Fiorentino (Firenze), Italy
**3** Department of Physics, Technion, Haifa 32000, Israel
* n.pinzanifokeeva@gmail.com

October 1, 2021

## Abstract

We construct an approximately conserved current for $2+1$ dimensional, Aristotelian (non boost invariant), fluid flow. When Aristotelian symmetry is enhanced to Galilean symmetry, this current matches the enstrophy current responsible for the inverse cascade in incompressible fluids. Other enhancements of Aristotelian symmetry discussed in this work include Lorentzian, Carrollian and Lifshitz scale symmetry.

# 1 Introduction

Enstrophy is an approximately conserved quantity in two dimensional, incompressible, fluid flow. Its existence is crucial for the appearance of the inverse energy cascade in two dimensional turbulence [1] and its dynamics is used in modeling turbulent dependent phenomena. While enstrophy is well understood in the context of incompressible fluid flow, barotropic flow and relativistic fluids (see [2–5]), little is known regarding its existence for more general flows with varying degrees of symmetry.

The goal of this work is to study enstrophy conservation for generic fluids in two spatial dimensions with little symmetry. More precisely, we will consider fluids which possess translation invariance in space and time, rotational invariance in space, and have a well defined thermodynamic limit. We will refer to these fluids as non frame invariant or Aristotelian fluids. The dynamics of such fluids is of interest due to its proximity to the dynamics of flocking behavior [6]. Also, by an appropriate enhancement of symmetries, the non frame invariant fluid equations of motion transform into a variety of boost invariant or Lifshitz invariant fluid equations of motion [7–10].

In incompressible flow, the enstrophy current can be written in the form

$$J^\mu_{\text{inc}} = \frac{\omega^2}{n} \left(1, \vec{v}\right) \tag{1}$$

with $\vec{v}$ the velocity field, $n$ the particle number density, assumed to be constant, and $\omega^2 = \omega_{ij}\delta^{ii'}\delta^{jj'}\omega_{i'j'}$ where

$$\omega_{ij} = \partial_i v_j - \partial_j v_i\,. \tag{2}$$

Here and in what follows Greek indices $\mu, \nu, \ldots$ denote spacetime coordinates while Latin indices $i, j, \ldots$ denote spatial coordinates. (Often, $n$, being constant, is omitted from (1).) It is straightforward to check that (1) is conserved under the Euler equations and is negative semidefinite once viscous corrections are introduced. In fact, there is not one, but an infinite set of conserved enstrophy currents,

$$J^\mu_{(\alpha)} = q\left(\frac{s}{n}\right)\frac{(\omega^2)^\alpha}{n^{2\alpha-1}}\left(1, \vec{v}\right)\,, \tag{3}$$

with $\alpha$ a real number, $s$ the entropy density and $q$ an arbitrary function. (Equivalently, one can combine powers of $J^\mu_{(\alpha)}$ into $J^\mu_Q = n\,Q(s/n, \omega^2/n^2)(1, \vec{v})$ with $Q$ a generic function of its arguments, and parameterize the enstrophy currents with the function $Q$.) These currents can be further generalized in the presence of additional conserved $U(1)$ charges, or if the flow is barotropic. See, e.g., [5]. For $\alpha = 1$ and $q = 1$ we obtain (1).

In this work we generalize (1) and show that a fluid flow whose dynamics is determined by space-time translation invariance, rotation invariance, and the requirement of a thermodynamic limit possesses a family of conserved enstrophy currents of the form

$$J^\mu = q\left(\frac{s}{n}\right)\frac{(\Omega^2)^\alpha}{s^{2\alpha-1}}u^\mu\,. \tag{4}$$

Here $u^\mu$ is an appropriate velocity field such that $su^\mu$ is the leading contribution to the entropy current and $nu^\mu$ is a conserved $U(1)$ current (which presumably exists). Also, $q$ is an arbitrary function of $s/n$ and $\Omega^2$ a generalized squared vorticity whose form depends on the underlying symmetries of the theory and whose value can be found in table 1 (and its detailed derivation in section 3). We have intentionally kept the normalization of the velocity field $u^\mu$ vague since it depends on the type of symmetries we intend to keep. See table 2 and section 3 for details. Curiously, $J^\mu$ with $\alpha = 0$ and $q = 1$ is the entropy current.

We claim that the current (4) is conserved either for a particular class of equations of state (summarized in table 2) which in the Galilean invariant limit reduce to the barotropic condition, or for a generalized incompressible flow in which case (4) reduces to (1) when Galilean invariance becomes a symmetry of the equations of motion.

Our work is organized as follows. In section 2 we identify sufficient conditions for the existence of an enstrophy current in a flat background: that there exists a closed two-form $\Omega$ satisfying $i_u\Omega = 0$ under the equations of motion (here $i_u$ is the interior product with the velocity vector $u^\mu$). In section 3 we solve the condition $i_u\Omega = 0$ for Aristotelian fluids by constraining the equation of state. In the same section we study various limits of the enstrophy current once boost or scale symmetries are present. Later, in section 4 we solve $i_u\Omega = 0$ in the incompressible limit. We end with a summary and brief discussion in section 5. A brief review of Galilean and Carrollian invariant hydrodynamics is relegated to the appendix.

# 2 A sufficient condition for the existence of an enstrophy current

In this work we will be interested in constructing an enstrophy current which is conserved in a flat background geometry. Since we are interested in dynamical systems which do not necessarily possess any type of boost invariance (Lorentz, Galilean or other), we start with a brief discussion of Aristotelian geometry which will help us ensure Aristotelian covariance. It will also help us to ensure that our theory is coordinate invariant and later, if the theory does possess boost invariance, to ensure compatibility of the isometries of space-time with the symmetries of the equations of motion. After a brief discussion of a non boost invariant geometry we will turn our attention to the construction of the enstrophy current. As stated earlier, we will then argue that the existence of a closed two-form $\Omega$ with $i_u\Omega = 0$ is sufficient to obtain an enstrophy current in a hydrodynamic setting.

Consider a manifold equipped with an inverse metric $h^{\mu\nu}$ which is degenerate in the sense that there exists an $n_\mu$ such that $h^{\mu\nu}n_\nu = 0$. The manifold is also equipped with a vector $\bar{n}^\mu$ which we refer to as the time direction and which we take, without loss of generality, to satisfy $n_\mu\bar{n}^\mu = -1$. We refer to $h^{\mu\nu}$ as the (inverse) spatial metric. In Cartesian coordinates, the flat metric and time direction are given by

$$h^{\mu\nu} = \delta_i^\mu \delta_j^\nu \delta^{ij}, \qquad \bar{n}^\mu = \delta^\mu{}_0. \tag{5}$$

The barred notation used in the previous paragraph is to emphasize that $\bar{n}^\mu$ is not obtained by raising the indices of $n_\mu$ with the spatial metric. However, from $h^{\mu\nu}$ and $\bar{n}^\mu$ we can construct $\gamma^{\mu\nu} = h^{\mu\nu} - \bar{n}^\mu\bar{n}^\nu$ and its inverse $\gamma_{\mu\nu}$ from which $\bar{n}^\mu = \gamma^{\mu\nu}n_\nu$. The tensor $\gamma^{\mu\nu}$ can also be used to define $\bar{h}_{\mu\nu} = \gamma_{\mu\nu} + n_\mu n_\nu$. Note that

$$\bar{n}^\mu\bar{h}_{\mu\nu} = 0, \qquad P^\mu{}_\nu \equiv h^{\mu\alpha}\bar{h}_{\alpha\nu} = \delta^\mu{}_\nu + \bar{n}^\mu n_\nu. \tag{6}$$

Since there is no preferred choice of metric or connection on the manifold, we will use the inverse metric

$$g_A^{\mu\nu} = h^{\mu\nu} - N\bar{n}^\mu\bar{n}^\nu \tag{7a}$$

to raise and lower indices. In an Aristotelian geometry we may take $N$ to be 1 or 0 without loss of generality. In the limit where the spacetime symmetry is enhanced to Lorentz invariance the metric must take the form

$$g_L^{\mu\nu} = g_A^{\mu\nu}\Big|_{N=1} = h^{\mu\nu} - \bar{n}^\mu\bar{n}^\nu. \tag{7b}$$

Likewise, the $N = 0$ metric corresponds to theories where the spacetime symmetry is enhanced to (massive) Galilean invariance,

$$g_G^{\mu\nu} = g_A^{\mu\nu}\Big|_{N=0} = h^{\mu\nu}. \tag{7c}$$

The somewhat peculiar Carrollian symmetry is associated with a metric

$$g_C^{\mu\nu} = h^{\mu\nu} - P_\alpha^\mu M_C^\alpha \bar{n}^\nu - P_\alpha^\nu M_C^\nu \bar{n}^\mu + \bar{n}^\mu \bar{n}^\nu M_C^2, \tag{7d}$$

where $M_C^\mu$ is an auxiliary background field necessary to ensure Carrollian invariance [11] and $M_C^2 = M_C^\alpha M_C^\beta \bar{h}_{\alpha\beta}$. We will discuss these enhanced symmetries in section 3 when they will be more relevant. We will often use $g^{\mu\nu}$ for the inverse metric to denote any one of (7).

In a Riemannian geometry there is a unique torsion free, metric compatible connection. This is not the case for Aristotelian, Galilean or Carrollian geometries. There, a well defined connection requires the introduction of additional fields of which $M_C^\mu$ introduced in the context of Carrollian geometries above is an example. Luckily, for the purpose of this work, we only need to ensure that an appropriate metric compatible connection exists. If $g^{\mu\nu}$ is invertible then we can use the Christoffel connection as our connection. For metrics of the form $g_A^{\mu\nu}\Big|_{N=0}$ we may use the Newton-Cartan connection

$$\Gamma^\mu{}_{\nu\rho} = -\bar{n}^\mu \partial_\nu n_\rho + \frac{1}{2} h^{\mu\sigma} \left( \partial_\nu \bar{h}_{\rho\sigma} + \partial_\rho \bar{h}_{\nu\sigma} - \partial_\sigma \bar{h}_{\nu\rho} \right), \tag{8}$$

which is compatible with $h^{\mu\nu}$ and $n_\mu$ and is torsion free as long as $d(n_\alpha dx^\alpha) = 0$. For the metric $g_C^{\mu\nu}$ we may use the same connection (8) appropriately modified to include $M_C^\mu$ terms, see [11], such that it is compatible with $g_C^{\mu\nu}$ and $n_\mu - \bar{h}_{\mu\alpha} M_C^\alpha$. We note that it is also possible to construct a connection which is compatible with $h^{\mu\nu}$ and $n_\mu$ and with $\bar{h}_{\mu\nu}$ and $\bar{n}^\mu$. See Appendix A for details. In what follows we will assume the background geometry to be torsion free.

A sufficient condition for

$$J_e^\mu = \frac{\Omega^2}{s} u^\mu \tag{9}$$

to be conserved is that

$$S^\mu = s u^\mu \tag{10}$$

is conserved ($\nabla_\mu S^\mu = 0$) and that

$$\Omega^2 = -W^\mu{}_\mu \tag{11}$$

with

$$W^\nu{}_\mu = \Omega^{\nu\alpha} \Omega_{\alpha\mu}, \qquad \Omega^{\alpha\beta} = g^{\alpha\mu} g^{\beta\nu} \Omega_{\mu\nu}, \tag{12}$$

such that

$$\Omega^{\mu\nu} \nabla_\alpha \left( u^\alpha \Omega_{\mu\nu} \right) = 0 \tag{13}$$

at least under the equations of motion.

Recall that the Galilean enstrophy current described in the introductory section is conserved only in the absence of viscous terms. Therefore, the current $J_e^\mu$ in (9) should be conserved in the absence of dissipative terms as well. In a hydrodynamic setting the entropy current is always conserved in the absence of dissipative terms and is therefore always available in order to construct (10) with $s$ the entropy density and $u^\mu$ the appropriate velocity field. In the presence of an additional conserved $U(1)$ charge, $N^\mu = n u^\mu + \ldots$, one can construct a broad class of enstrophy currents of the form (4) which are conserved due

to the conservation laws for $S^\mu$, $N^\mu$ and equation (13). In the remainder of this section we will describe an operative algorithm for constructing a two-form $\Omega_{\mu\nu}$ which satisfies (13). Indeed, as we will now show, in order to maintain (13) in a flat background geometry, it is sufficient to require that $\Omega$ is closed and orthogonal to $u^\mu$

$$d\Omega = 0\,, \tag{14a}$$

$$u^\alpha\Omega_{\alpha\beta} = 0\,, \tag{14b}$$

at least under the equations of motion.

Given (14) we have $\pounds_u\Omega_{\mu\nu} = 0$ with $\pounds_u$ the Lie derivative in the $u$ direction. Thus,

$$\nabla_\alpha\left(u^\alpha\Omega_{\mu\nu}\right) = \Omega_{\mu\nu}\nabla_\alpha u^\alpha - \Omega_{\alpha\nu}\nabla_\mu u^\alpha - \Omega_{\mu\alpha}\nabla_\nu u^\alpha\,. \tag{15}$$

It is convenient to decompose $\nabla_\mu u^\alpha$ into components which are parallel and perpendicular to vectors $\bar\tau^\alpha$ and $\tau_\alpha$ which satisfy $\tau_\mu\bar\tau^\mu = -1$, viz.,

$$\nabla_\mu u^\alpha = \tau_\mu\bar\tau^\alpha S + P^\alpha_{(\tau)\beta}\tau_\mu j^\beta + \bar\tau^\alpha P^\gamma_{(\tau)\mu}\bar j_\gamma + \frac{1}{d}P^\alpha_{(\tau)\mu}\Theta + \Sigma^\alpha{}_\mu\,, \tag{16}$$

where

$$P^\alpha_{(\tau)\beta} = \delta^\alpha{}_\beta + \bar\tau^\alpha\tau_\beta\,, \tag{17}$$

$d$ is the number of spatial dimensions, $\Sigma^\alpha{}_\mu\bar\tau^\mu = \Sigma^\alpha{}_\mu\tau_\alpha = 0$, and $\Sigma^\alpha{}_\alpha = 0$. A straightforward computation yields

$$\Omega^{\mu\nu}\nabla_\alpha\left(u^\alpha\Omega_{\mu\nu}\right) = S\left(2\tau_\alpha W^\alpha{}_\beta\bar\tau^\beta - \Omega^2\right) + \frac{1}{d}\Theta\left(2\tau_\alpha W^\alpha{}_\beta\bar\tau^\beta + (d-2)\Omega^2\right)$$
$$+ 2\left(\Sigma^\alpha{}_\beta W^\beta{}_\alpha + \bar j_\alpha P^\alpha_{(\tau)\beta}W^\beta{}_\gamma\bar\tau^\gamma + \tau_\alpha W^\alpha{}_\beta P^\beta_{(\tau)\gamma}j^\gamma\right)\,. \tag{18}$$

In order for (13) to hold, the right-hand-side of (18) must vanish under the equations of motion. Let us consider each such term separately. To ensure that $W^\alpha{}_\beta\bar\tau^\beta$ vanishes, it is convenient to choose $\bar\tau^\mu \propto u^\mu$ so that $W^\alpha{}_\beta\bar\tau^\beta = 0$ as a result of $\Omega_{\alpha\beta}u^\beta = 0$. The simplest method by which we can set $\tau_\alpha W^\alpha{}_\beta = 0$ is to use a non invertible $g^{\mu\nu}$ and then choose $\tau_\alpha$ as the (appropriately normalized) eigenvector of $g^{\mu\nu}$ with zero eigenvalue, $g^{\mu\nu}\tau_\nu = 0$. If we are insistent on choosing an invertible metric $g^{\mu\nu}$ then setting $\tau_\alpha = g_{\alpha\nu}u^\nu$ will ensure that $\tau_\alpha W^\alpha{}_\beta = 0$. Such a construction is viable only if

$$g_{\mu\nu}u^\mu u^\nu = \bar h_{\mu\nu}u^\mu u^\nu - Nn_\mu n_\nu u^\mu u^\nu \tag{19}$$

has definite sign. Otherwise, some modifications to the normalization $\tau_\alpha\bar\tau^\alpha = -1$ may be needed. Either way, (18) reduces to

$$\Omega^{\mu\nu}\nabla_\alpha\left(u^\alpha\Omega_{\mu\nu}\right) = -S\Omega^2 + \Theta\left(1 - \frac{2}{d}\right)\Omega^2 + 2\Sigma^\alpha{}_\beta W^\beta{}_\alpha\,. \tag{20}$$

The second term on the right of (20) vanishes in $2 + 1$ spacetime dimensions ($d = 2$). The last term also vanishes for the following reason. Consider the tensor $\Omega^\alpha{}_\beta = g^{\alpha\mu}\Omega_{\mu\beta}$. By construction it has one zero eigenvalue associated with the eigenvector $\bar\tau^\beta$. Let us denote another eigenvector by $p^\alpha$. Skew symmetric matrices have either zero or imaginary eigenvalues and their rank is even. Thus, since $\Omega_{\mu\nu}$ is non vanishing, we must have $\Omega^\alpha{}_\beta p^\beta = icp^\alpha$ with $c \neq 0$ and $c \in \mathbb{R}$, and the third eigenvector of $\Omega^\alpha{}_\beta$ is $(p^\beta)^*$. Evaluating $\Omega^\alpha{}_\beta$ and then $W^\alpha{}_\beta$ in terms of this orthogonal basis, it is straightforward to show that $W^\alpha{}_\beta$ is proportional to $P^\alpha_{(\tau)\beta}$. It follows that

$$\Sigma^\alpha{}_\beta W^\beta{}_\alpha = 0 \tag{21}$$

on account of tracelessness of $\Sigma^\alpha{}_\beta$. Note that (21) is referred to as the absence of vortex stretching in the context of incompressible fluid flow. See, e.g., [12].

Using (16), the coefficient of the first term on the right-hand-side of (20) evaluates to

$$S = \bar{\tau}^\mu \tau_\alpha \nabla_\mu u^\alpha \,. \tag{22}$$

There are several instances under which $S$ vanishes. If we use an invertible $g^{\alpha\beta}$ (so that $\tau_\alpha \propto g_{\alpha\beta} u^\beta$) and we also have $u^\mu g_{\mu\nu} u^\nu = c_0$ with $u^\mu$ (and the constant, $c_0$) real, then we obtain $S = 0$. Otherwise, we end up with the equivalent constraints

$$u^\mu \tau_\alpha \nabla_\mu u^\alpha = 0 \qquad \text{or} \qquad u^\mu u^\alpha \nabla_\mu \tau_\alpha = 0 \,. \tag{23}$$

Thus, the right-hand-side of (20) vanishes when the geometry is such that $\tau_\alpha$ is covariantly constant, or when we choose a geometry which is compatible with the dynamics of $u^\alpha$ in such a way that $\tau_\alpha \nabla_\mu u^\alpha = 0$. Notice that, in a flat background, $n_\mu = -\delta^0_\mu$ in Cartesian coordinates, which is covariantly constant.

To summarize, we have shown that equation (14) is sufficient to ensure (13) as long as $g^{\mu\nu}$ is invertible and we can choose $u^\mu g_{\mu\nu} u^\nu = c_0$ with real $u^\mu$. Or, one of (23) is satisfied. It now remains to compute under which conditions (14) is satisfied. We will solve this condition in two instances. In section 3 we will look for the most general (first order in derivatives) $\Omega_{\mu\nu}$ and equation of state such that (14) is satisfied. In section 4 we will take the incompressible (low Mach number) limit of the fluid equations and look for a solution which does not rely on a particular class of equations of state.

## 3 Solving $i_u \Omega = 0$ by constraining the equation of state

Our task is now reduced to constructing an $\Omega$ satisfying (14) at least under the equations of motion. We will be interested in solutions to the equations of motion in a flat background where $S$ in (22) vanishes and in the absence of any external forces. In such a background, the equations of motion of an inviscid fluid in the absence of any type of boost symmetry are given by [7],

$$
\begin{aligned}
0 &= \partial_0 v_i + v^k \partial_k v_i + \frac{1}{\rho} \partial_i P + \frac{v_i}{\rho} \left( \partial_0 \rho + \partial_k \left( v^k \rho \right) \right) \,, \\
0 &= \partial_0 s + \partial_i \left( s v^i \right) \,, \\
0 &= \partial_0 n + \partial_i \left( n v^i \right) \,.
\end{aligned}
\tag{24}
$$

Here $v^i$ is a velocity field as seen in the lab frame, $s$, $n$ and $\rho$ are the entropy density, a conserved $U(1)$ charge density, and the kinetic mass density respectively. The latter is related to the momentum density covector $P_i$ via $P_i = \rho \delta_{ij} v^j$. The entropy density $s$ is conjugate to the temperature $T$ and the charge density $n$ is conjugate to a chemical potential $\mu$. The pressure, $P$, is related to the remaining hydrodynamic variables via

$$dP = s\,dT + n\,d\mu + \frac{1}{2}\rho\,dv^2 \,. \tag{25}$$

We emphasize that viscous terms are absent from (24). An analysis of viscous corrections to (24) can be found in [8–10]. Since we are interested in an approximate conservation law for the enstrophy, valid when viscous terms are negligible, it is sufficient to consider (24). A full derivation of (24) can be found in [7]. In brief, translation invariance implies the existence of a stress tensor. Insisting that in thermodynamic equilibrium the energy

flux, momentum flux and charge flux move with the same velocity, and that the free energy can be identified with the pressure leads to (24). It is interesting to contrast (24) with the equations of motion in [6] valid under the same symmetry group but in the absence of thermodynamic equilibrium.

It is convenient to define the velocity vectors $v^\mu$ and $\bar{v}_\mu$ satisfying

$$\bar{v}_\mu = \bar{h}_{\mu\nu} v^\nu \,, \qquad v^\mu n_\mu = -1 \,, \tag{26}$$

(so that

$$v^\mu = (1,\, v^i) \,, \qquad \bar{v}_\mu = (0,\, v_i) \,, \tag{27}$$

in the coordinate system (5)) and

$$\Theta = \nabla_\mu v^\mu \,, \qquad a_\mu = v^\alpha \nabla_\alpha \bar{v}_\mu \,, \qquad \tilde{n} = n/s \,, \tag{28}$$

and split the derivatives into transverse and parallel components such that

$$\nabla_\nu = D_\nu^\perp - n_\nu D \,, \qquad \bar{n}^\mu D_\mu^\perp = 0 \,. \tag{29}$$

In this notation the equations (24) take the form

$$\begin{aligned} 0 &= a_\mu + \frac{1}{\rho} D_\mu^\perp P + \frac{\bar{v}_\mu}{\rho} \left( \rho\Theta + v^\nu \partial_\nu \rho \right) \,, \\ 0 &= D\tilde{n} + v^\mu D_\mu^\perp \tilde{n} \,, \\ 0 &= Ds + s\Theta + v^\mu D_\mu^\perp s \,. \end{aligned} \tag{30}$$

Note that $v^\mu a_\mu = \frac{1}{2} v^\nu \partial_\nu v^2$ with $v^2 = v^\mu v^\nu \bar{h}_{\mu\nu}$ and that $\bar{n}^\mu a_\mu = 0$.

We have made a distinction between the velocity $v^\mu$ which we have introduced in (27) and the velocity $u^\mu$ that appears in, say, (10). In the later part of this work we will find that the natural velocity field $u^\mu$ which appears in thermodynamic quantities, as in, e.g., (10), is proportional to, if not equal to, $v^\mu$. (For instance, in a Lorentz invariant system $u^\mu = v^\mu / \sqrt{1 - v^2}$ while in a Galilean invariant system it is natural to use $u^\mu = v^\mu$.)

Recall that our goal is to find an $\Omega$ that satisfies (14) under the equations of motion. To this end, let us start with the ansatz

$$\Omega = d(f\bar{v}) + d(gn) \,, \tag{31}$$

where $\bar{v} = \bar{v}_\mu dx^\mu$, $n = n_\mu dx^\mu$, and $f$ and $g$ are arbitrary functions of $s$, $\tilde{n} = n/s$ and $v^2$.[1] In the presence of an external gauge field we could add to (31) its associated field strength. Equation (31) ensures that $\Omega$ is closed. It remains to find the constraints on the pressure and on $f$ and $g$ so that $\Omega_{\mu\nu} u^\nu = 0$ under the equations of motion. Under the assumption that $u^\mu \propto v^\mu$ this amounts to solving $\Omega_{\mu\nu} v^\nu = 0$.

The computation we wish to carry out is now straightforward though somewhat technical. In what follows we will highlight its salient features. First, we found it convenient to replace the pressure $P$ with the potential $G$,

$$G + P = sT + \mu n \tag{32}$$

in which case we have

$$dG = T ds + n d\mu - \frac{1}{2} \rho dv^2 \,. \tag{33}$$

---

[1] It is somewhat unfortunate that we have used both $n = n_\mu dx^\mu$ and $n$ the charge density in the same sentence. We hope that this notation does not confuse the reader and that the appropriate choice of $n$ is clear from context.

Next, we note that the equation of motion for $a_\mu$ in (30) can be dotted with $v^\mu$ to obtain

$$\Theta v^2 + \frac{1}{2}Dv^2 + \frac{v^2}{\rho}D\rho + \frac{1}{\rho}v^\alpha D_\alpha^\perp P + \frac{1}{2}v^\alpha D_\alpha^\perp v^2 + \frac{v^2}{\rho}v^\alpha D_\alpha^\perp \rho = 0. \tag{34}$$

Thus, we can use the equations of motion (30) to get rid of $a_\mu$, $Dv^2$, $D\tilde{n}$ and $Ds$. Since $Dv^2$ drops out of (34) if $\partial_{v^2}G + 2v^2\partial_{v^2}^2 G = 0$, we will focus, for now, on generic expressions for $G$ and treat $\partial_{v^2}G + 2v^2\partial_{v^2}^2 G = 0$ as a special case.

After some algebra, it is possible to show that, under the equations of motion

$$\Omega_{\mu\nu}v^\nu = \sum_a B_\mu^a \beta_a \tag{35}$$

where

$$B_\mu^1 = (v^2 n_\mu + \bar{v}_\mu)v \cdot D^\perp s, \quad B_\mu^2 = (v^2 n_\mu + \bar{v}_\mu)v \cdot D^\perp \tilde{n}, \quad B_\mu^3 = (v^2 n_\mu + \bar{v}_\mu)v \cdot D^\perp v^2,$$
$$B_\mu^4 = (v^2 n_\mu + \bar{v}_\mu)\nabla \cdot v, \qquad B_\mu^5 = n_\mu v \cdot D^\perp s + D_\mu^\perp s, \qquad B_\mu^6 = n_\mu v \cdot D^\perp \tilde{n} + D_\mu^\perp \tilde{n},$$
$$B_\mu^7 = n_\mu v \cdot D^\perp v^2 + D_\mu^\perp v^2. \tag{36}$$

Thus, $\Omega_{\mu\nu}v^\mu = 0$ reduces to $\beta_a = 0$. Our strategy for solving these equations is as follows. First we solve $\beta_a = 0$ with $a = 5, \dots, 7$ algebraically for $\partial_s g$, $\partial_{\tilde{n}} g$ and $\partial_{v^2} g$. We then construct from the above solutions additional equations $\kappa_1 = 0$, $\kappa_2 = 0$ and $\kappa_3 = 0$ by requiring that mixed partial derivatives of $g$ are compatible.

Assuming that $f \neq 0$ (since if $f = 0$ implies that $g$ is constant in which case we get the trivial solution on account of the torsionless condition $\partial_\mu n_\nu - \partial_\nu n_\mu = 0$) we find from $\beta_1 = 0$ that

$$f = f_1(s, \tilde{n})\partial_{v^2}G \qquad \text{or} \qquad G = G_1(s, \tilde{n}) + sG_2(\tilde{n}, v^2). \tag{37}$$

We refer to these two branches of solutions as branch A and branch B respectively. For branch A one finds that the above solution automatically sets $\beta_2 = \beta_3 = 0$. One can then solve $\beta_4 = 0$ which gives

$$f_1 = \frac{1}{s\xi(\tilde{n})}, \tag{38}$$

with $\xi$ an arbitrary function of $\tilde{n}$. The remaining non trivial equations are $\kappa_2 = 0$ and $\kappa_3 = 0$. The solution to the former is

$$G = H\left(s\xi(\tilde{n}), v^2\right) + \eta(s, \tilde{n}), \tag{39}$$

with $H$ an arbitrary function of its two variables. The solution to the remaining $\kappa_3 = 0$ is

$$\eta = \eta_1\left(s\xi(\tilde{n})\right) + s\eta_2(\tilde{n}). \tag{40}$$

It is then straightforward to go back and solve $\beta_a = 0$ for $a = 5, \dots, 7$ for $g$ and obtain (after some relabeling)

$$f = \frac{\partial_{v^2}H}{s\xi}, \qquad g = \frac{v^2}{s\xi}\partial_{v^2}H - \frac{1}{2}\partial_{s\xi}H, \qquad G = H + s\eta, \tag{41}$$

where the arguments of the various functions are

$$H = H(s\xi, v^2), \qquad \xi = \xi(\tilde{n}), \qquad \eta = \eta(\tilde{n}), \tag{42}$$

and we have removed a constant, $g_0$, from $g$ since it does not contribute to $\Omega$. Notice that in this case we have

$$P = -H + s\xi\partial_{s\xi}H. \tag{43}$$

Since $s\eta_2$ is not absolutely convex it does not contribute to the pressure.

The strategy for solving the B branch is similar to that of the A branch. One finds that the B branch splits into two branches. One of them is a special case of the A branch solution. The other is given by

$$f = f(v^2), \qquad g = v^2 f(v^2) - \frac{1}{2} \int^{v^2} f(x) dx, \qquad G = -P_0 + sJ(\tilde{n}, v^2). \qquad (44)$$

In this case we find that

$$P = P_0. \qquad (45)$$

It remains to treat the special case $\partial_{v^2} G + 2v^2 \partial_{v^2}^2 G = 0$ in which case $Dv^2$ becomes an independent variable. This case can be solved by the same method as the generic case and one finds that it leads to several branches of solutions all of which coincide with or are a special case of (41). At the end of the day, we find that (41) is valid for any $H$ as long as $\xi \neq 0$ and $\partial_{v^2} H \neq 0$ and (44) is valid whenever $\partial_{v^2} G + 2v^2 \partial_{v^2}^2 G \neq 0$.

In the absence of additional symmetries we have from (24) and (10) that $u^\mu = v^\mu$. In this case, for general values of $N$ in (7a) we find

$$u^\mu g_{\mu\nu} u^\nu = N - v^2. \qquad (46)$$

Since the right-hand-side of (46) is not sign definite we can not enforce (23) and then the resulting enstrophy current can not be conserved.[2] Thus, we are forced to set $N = 0$ in order to obtain a non trivial enstrophy current. The resulting enstrophy current is then given by (4) with

$$\Omega^2 = \Omega_{\mu\nu} \Omega_{\rho\sigma} h^{\mu\rho} h^{\nu\sigma}, \qquad (47)$$

and $u^\mu = v^\mu$. Note that the contributions to $\Omega$ coming from $g$ in (31) drop off from the expression for $\Omega^2$ due to the non torsion condition $dn = 0$ and $h^{\mu\nu} n_\nu = 0$. We have summarized our results in tables 1 and 2.

While our result for the enstrophy current is very general, it is interesting to study its behavior in several limiting cases where the Aristotelian symmetry is lifted to one with some boost invariance. In particular, following [7], we will be interested in equations of state where Lorentz invariance, Galilean invariance, Carrollian invariance or Lifshitz invariance are present. We now turn our attention to these non generic cases.

## 3.1 Recovering the Galilean invariant solution

We can use our generic results (41) or (44) to construct a Galilean covariant enstrophy current so long as we restrict the above solutions to those which possess Galilean symmetry and also ensure that the resulting expression for the enstrophy current transforms covariantly under Galilean boosts.

Recall that the Galilean group is generated by the (massive) Bargmann algebra. To enhance the generic fluid equations of motion to those of a Galilean invariant fluid [13–18], one has to relate the kinematic mass term $\rho$ to the mass density of the fluid [7]

$$\rho = mn \qquad (48)$$

---

[2]Note that we may also attempt the following: in place of (10) we can use $S^\mu = \sigma u^\mu$ where $u^\mu$ is chosen such that $u^\mu \propto v^\mu$ with a proportionality constant such that $u^\mu g_{\mu\nu} u^\nu = c_0$. With this choice of $u^\mu$ we will get $S = 0$ in (22) and so, should be able to construct an enstrophy current as in (4) with $s$ replaced by $\sigma$ (and $n$ replaced by $\nu$ with $N^\mu = \nu u^\mu$). The problem with this construction is that there may exist solutions where locally $v^2 > N$ and also $v^2 < N$. For such configurations, if we normalize the velocity $u^\mu$ to take a constant value then we will end up with a complex valued velocity field, c.f., (46). When we will discuss Lorentzian symmetry we will see that the a construction of the type described in this footnote is viable.

where $n$ the particle number density and $m$ is the mass.

For the solution in (41) the identification given in (48) implies

$$-2\partial_v^2 H(s\xi,\, v^2) = mn\,. \tag{49}$$

This is a differential equation for both $H$ and $\xi$ and can be solved by integrating over $v^2$ and expanding in a power series in $s$ and $\tilde{n}$. We find

$$H = H_G(s\tilde{n}) - \frac{1}{2}ms\tilde{n}v^2\,, \qquad \xi = -\frac{m\tilde{n}}{2f_0}\,, \tag{50}$$

which gives us

$$f = f_0\,, \qquad g = \frac{1}{2}f_0 v^2 + \frac{f_0}{m}H_G'\,, \qquad G = H_G(s\tilde{n}) + s\eta(\tilde{n}) - \frac{1}{2}ms\tilde{n}v^2\,, \tag{51}$$

and a barotropic pressure term

$$P(n) = -H_G(n) + nH_G'(n)\,. \tag{52}$$

The result (52) leads to the known approximately conserved enstrophy current which exists in compressible barotropic flow.

Solving (48) for (44) yields

$$f = f(v^2)\,, \qquad g = v^2 f(v^2) - \frac{1}{2}\int^{v^2} f(x)dx\,, \qquad G = -P_0 + sJ_G(\tilde{n}) - \frac{1}{2}mnv^2\,. \tag{53}$$

We will see shortly that while (53) solves (49) and (14) it does not allow for a Galilean covariant enstrophy current.

In order to construct a Galilean covariant enstrophy current we need to identify a Galilean covariant velocity field, $u_G^\mu$, and ensure that $\Omega^2$ is a scalar under Galilean boosts. Let us start with the former. The natural velocity field to use in a Galilean invariant theory is one which transforms covariantly under a change of reference frame. By this we mean the following: if $u_G^\mu(\vec{v})$ specifies the velocity of a particle moving with velocity $\vec{v}$, then it must be the case that when we transform to a coordinate system moving at constant velocity $\vec{v}_0$ relative to the first,

$$G^\mu{}_\nu(\vec{v}_0)u_G^\nu(\vec{v}) = u_G^\nu(\vec{v} + \vec{v}_0)\,, \tag{54}$$

with $G^\mu{}_\nu(\vec{v}_0)$ representing a Galilean boost to the reference frame moving at velocity $\vec{v}_0$. Equation (54) implies that

$$u_G^\mu = v^\mu\,. \tag{55}$$

Of course, (55) could have been obtained by considering the change in the particles coordinates $X^\mu(\tau)$ relative to the Galilean invariant proper time, or by taking the small velocity limit of a relativistic velocity field.

Next, consider $\Omega^2 = h^{\mu\alpha}h^{\nu\beta}\Omega_{\mu\nu}\Omega_{\alpha\beta}$. If $\Omega^2$ is a scalar it should be invariant under Galilean transformations. Recall that a Galilean transformation on dynamical fields is a coordinate transformation of the type (104) while a Galilean transformation on the background fields $\bar{h}_{\mu\nu}$ and $\bar{n}^\mu$ involves, in addition, a Milne transformation. The latter ensures that, as opposed to the situation in an Aristotelian geometry, there is an equivalence class of time directions: $\bar{n}_1^\mu \sim \bar{n}_2^\mu$, if $n_\mu \bar{n}_1^\mu = n_\mu \bar{n}_2^\mu = -1$. Thus, in particular, for a flat background geometry, we find that

$$\bar{h}_{\mu\nu} \xrightarrow[Galilean]{} \bar{h}_{\mu\nu} + 2\lambda^2 n_\mu n_\nu \tag{56}$$

where $\lambda_\mu$ is the boost parameter (e.g., in Cartesian coordinates $\lambda_\mu = (0, \vec{v}_0)$) and $\lambda^2 = \lambda_\mu \lambda_\nu h^{\mu\nu}$. See appendix A for a concise summary or [14, 19–25] for an extended discussion. Thus, while $u_G^\mu = v^\mu$ transforms covariantly under Galilean boosts, $\bar{u}_{G\,\mu} = \bar{v}_\mu = \bar{h}_{\mu\nu} v^\nu$ does not, implying that $\Omega_{\mu\nu}$ is not Galilean covariant.

A resolution to a problem of this type can be found in [14]. In Galilean invariant theories the Christoffel connection is not the unique, symmetric, metric compatible one. Rather, in order to define such a connection one needs, at the very least, to introduce an additional one form $A_\mu^G dx^\mu$. This one form is often identified with the gauge field associated with the inherent $U(1)$ symmetry which leads to mass conservation in Galilean theories. Metric compatibility then implies that $A_\mu^G$ does not transform covariantly under Galilean transformations. Despite that, in [14] it was shown that gauge invariant combinations constructed out of $A_\mu^G + \bar{h}_{\mu\nu} u_G^\nu + \frac{1}{2} n_\mu v^2$ are Galilean covariant.

So far, we have considered vanishing external sources for both the stress tensor and conserved $U(1)$ currents, and we may continue to do so by choosing a flat connection. The discussion in the previous paragraph suggests that in a Galilean invariant theory, (31) must be replaced by

$$\Omega = d\left( f^G \left( \bar{v} + A^G + \frac{1}{2} n v^2 \right) \right) + d(g^G n), \tag{57}$$

which forces $f^G = f_0$, a constant, on account of gauge invariance. (Recall that $f^G$ and $g^G$ are functions of the entropy density, charge density and velocity field and that $n = n_\alpha dx^\alpha$.) Since we can always choose a gauge where $A^G = 0$, we are free to use our generic results as long as we restrict $f$ to be a constant. Surprisingly, this is precisely the solution given in (51) with $g = \frac{1}{2} f_0 v^2 + g^G$ and $g^G = f_0 H_G'/m$. One can also choose $f$ to be a constant in (53) in which case (53) becomes a special case of (51).

Let us summarize our findings. A Galilean equation of state implies that (51) or (53) are valid expressions for constructing $\Omega_{\mu\nu}$. In order for $\Omega_{\mu\nu}$ to be Galilean covariant we must use $f = f_0$ in which case (53) becomes a special case of (51). In order to construct $\Omega^2$ we use the Galilean invariant metric $h^{\mu\nu}$ (or $g_A^{\mu\nu}$ with $N = 0$) which yields (47). $\Omega^2$ is then a scalar on account that the transformation of $\Omega_{\mu\nu}$ under Galilean boosts is proportional to $n_\mu$, which therefore vanishes when contracted with $h^{\mu\nu}$. Finally Galilean invariance also enforces that the velocity field in (4) is given by $u_G^\mu$ with (55).

## 3.2 Recovering the Lorentz invariant solution

Following [7], the fluid equations given in (24) are Lorentz covariant whenever

$$\rho = \frac{sT + \mu n}{1 - v^2}. \tag{58}$$

Inserting the first branch of solutions, (41), into (58) we find that (41) is restricted to take the form

$$H = H_L \left( \frac{s\xi}{\gamma} \right) - s\xi, \qquad \eta = \xi, \tag{59}$$

with $\gamma = 1/\sqrt{1 - v^2}$. In terms of $f$, $g$ and $G$ the first branch of solutions becomes (after some relabeling)

$$f = -\frac{1}{2} \gamma H_L'(s_L \xi(\tilde{n})), \qquad g = f, \qquad G = H_L(s_L \xi(\tilde{n})), \tag{60}$$

where we have defined

$$s_L = s/\gamma \tag{61}$$

and we note that we may write

$$\tilde{n} = \frac{n_L}{s_L}. \tag{62}$$

Indeed, as shown in [7] and as we will see shortly, $s_L$ and $n_L = n/\gamma$ are the relativistic expressions for the entropy density and $U(1)$ charge.

Going to the second branch of solutions, (44), we find that it takes the form

$$f = f(v^2), \qquad g = v^2 f(v^2) - \frac{1}{2} \int^{v^2} f(x) dx, \qquad G = -P_0 + s_L J_L(\tilde{n}), \tag{63}$$

under (58).

The Lorentz invariant metric is the Minkowski metric, or $g_{\mu\nu}$ with $N = 1$. The natural velocity field for a Lorentz invariant theory is given by

$$u_L^\mu = \gamma v^\mu \tag{64}$$

in which case the inviscid entropy current and charge current take the form $S^\mu = s_L u_L^\mu$ and $N^\mu = n_L u_L^\mu$ respectively. Clearly

$$\bar{u}_{L\,\mu} = g_{\mu\nu} u_L^\mu = \gamma \bar{v}_\mu + \gamma n_\mu \tag{65}$$

is also Lorentz covariant. Thus, the expression for $\Omega_{\mu\nu}$ given in (31), should reduce to

$$\Omega = d\left(f_L \bar{u}_L\right) \tag{66}$$

with

$$f_L = f/\gamma, \tag{67}$$

(and $\bar{u}_L = \bar{u}_{L\,\mu} dx^\mu$) if it is to be Lorentz covariant. The second branch of solutions, (63), does not meet this criterion (unless $f_L$ is constant) but the first branch of solutions, (60), does. Thus, $\Omega_{\mu\nu}$ takes the form given in (66) with (67) and (60). To construct (4) we use

$$\Omega^2 = \Omega_{\mu\nu} \Omega_{\rho\sigma} g_L^{\mu\rho} g_L^{\nu\sigma} \tag{68}$$

with $g_L^{\mu\nu}$ given in (7b) and $u^\mu = u_L^\mu$ given in (64). In this case, $g_{\mu\nu} u_L^\mu u_L^\nu = -1$ and therefore $S$ in (22) clearly vanishes.

In a relativistic setting it is often convenient to work with temperature and chemical potential instead of entropy density and charge density,

$$T\gamma = T_L = \frac{\partial G}{\partial s_L}, \qquad \mu\gamma = \mu_L = \frac{\partial G}{\partial n_L}. \tag{69}$$

It is straightforward to show that

$$\frac{T_L}{\mu_L} = -\tilde{n} + \frac{\xi(\tilde{n})}{\xi'(\tilde{n})}, \tag{70}$$

implying that $\tilde{n}$ is a function of $T_L/\mu_L$. Further, if we decompose

$$f_L = T_L f_r, \tag{71}$$

then

$$f_r = -\frac{1}{2\xi - \tilde{n}\xi'} \tag{72}$$

is also a function of $T_L/\mu_L$.

To obtain a simple expression for the pressure consider

$$H_L(x) = \int^x \frac{P(Q(-2y))}{y^2} dy \, , \tag{73}$$

where $Q(P'(x)) = x$. Note that $P = -G + s_L T_L + \mu_L s_L \tilde{n}$ is the pressure. In these variables we find that

$$Q(s\xi) = T_L f_r \tag{74}$$

implying

$$P = P\left(T_L f_r \left(\frac{T_L}{\mu_L}\right)\right) \tag{75}$$

which matches earlier results obtained for relativistic fluids [2, 5].

## 3.3 Carrollian symmetry

Carrollian invariance [26, 27] is perhaps the least familiar form of boost invariance. The Carrollian algebra can be obtained by taking the $c \to 0$ limit of the Lorentzian algebra. That is, it describes dynamics when the lightcone degenerates to a line. The most natural way to interpret this limit is by considering the limiting case of Lorentz transformations whose velocity parameter is much larger than the speed of light. To be explicit, consider

$$t \to t' = \frac{t - \frac{\vec{\beta} \cdot \vec{x}}{c}}{\sqrt{1 - \beta^2}} \, , \qquad \vec{x} \to \vec{x}' = \frac{\vec{x} - \vec{\beta} c t}{\sqrt{1 - \beta^2}} \, . \tag{76}$$

Often, we attribute these transformations to the dynamics of a massive particle: By identifying the velocity of the particle with the boost parameter required to bring it to a reference frame where it is stationary (in space) we obtain $\vec{\beta} = \vec{v}/c$. Following [27], we may use the same technique to identify $\vec{\beta} = c\vec{v}/v^2$ for tachyonic particles by equating the velocity of the tachyon with the boost parameter required to bring it to a reference frame where it is stationary in time. Taking the $c/|\vec{v}| \to 0$ limit of these transformations leads to Carrollian boosts

$$t' = t - \frac{\vec{v}}{v^2} \cdot \vec{x} \, , \qquad \vec{x}' = \vec{x} \, . \tag{77}$$

We comment that it is also possible to take the $c \to 0$ limit of the Lorentz transformations associated with subluminal velocities by scaling the velocity with $c^2$ ($\vec{v} \to c^2 \vec{v}/v^2$) as $c$ is taken to zero resulting also in (77). See, e.g., [28, 29]. This limit is potentially associated with the dynamics of massive particles trapped inside the lightcone that has shrunk to the $t$ axis. We also note that, curiously, the Carrollian algebra allows for a particular type of central extension in $2 + 1$ dimensions, [30], whose implications on hydrodynamics has not been worked out, at least as far as we know. It would be interesting to see whether this central charge relates to enstrophy.[3] We leave this direction to future work.

In order to have Carrollian covariant fluid equations we must set (see [7])

$$\rho = -\frac{sT + \mu n}{v^2} \, . \tag{78}$$

The first branch of solutions (41) now reads

$$G = H_C(s_C \xi(\tilde{n})) \, , \qquad f = \frac{1}{2} \frac{H'_C(s_C \xi(\tilde{n}))}{\sqrt{v^2}} \, , \qquad g = 0 \, , \tag{79}$$

---

[3]We thank W. Sybesma for pointing this out to us.

and the second branch of solutions (44) is given by

$$G = -P_0 + s_C J_C(\tilde{n}), \qquad f = f(v^2), \qquad g = v^2 f(v^2) - \frac{1}{2} \int^{v^2} f(x) dx. \qquad (80)$$

where $s_C = \sqrt{v^2} s$. (Note that (80) satisfies $\rho + 2v^2 \partial_{v^2} \rho = 0$ even though we assumed that it should not vanish. Nevertheless, it is still a solution to $i_u \Omega = 0$.)

As was the case for the Galilean theory, to ensure that the enstrophy current is Carrollian covariant we must identify a Carrollian covariant velocity field $u_C^\mu$ and show that $\Omega^2$ behaves as a scalar under Carrollian transformations. There are many equivalent ways of constructing a Carrollian covariant velocity field. Following the discussion for the Galilean covariant theory, we will determine $u_C^\mu$ by requiring that it is compatible with Carrollian addition of velocities.

$$\vec{v}' = \frac{\vec{v}}{1 - \vec{v} \cdot \frac{\vec{v}_0}{|\vec{v}_0^2|}}, \qquad (81)$$

where $\vec{v}_0$ is the boost parameter of the Carrollian transformation. It is now straightforward to show that

$$u_C^\mu(v_i) = \frac{1}{\sqrt{v^2}} \left(1, v^i\right) = \frac{v^\mu}{\sqrt{v^2}} \qquad (82)$$

is the unique vector that satisfies

$$C^\mu{}_\nu(\vec{v}_0) u_C^\nu(\vec{v}) = u_C^\mu(\vec{v}'), \qquad (83)$$

with $C^\mu{}_\nu$ a Carrollian transformation with boost parameter $\vec{v}_0$. (The same result can be obtained by considering the $c \to 0$ limit of the Lorentz invariant $u_L^\mu = \frac{\partial X^\mu}{\partial \tau}$ where $d\tau = \sqrt{-c^2 dt^2 + |d\vec{x}|^2}$). It follows that

$$\bar{u}_{C\,\mu} = \bar{h}_{\mu\nu} u_C^\nu = \left(0, \frac{v_i}{\sqrt{v^2}}\right) = \frac{\bar{v}_\mu}{\sqrt{v^2}} \qquad (84)$$

transforms covariantly under Carrollian boosts.

To determine whether $\Omega^2$ is a Carrollian scalar we must first determine the Carrollian transformation laws of the geometric data $\bar{h}_{\mu\nu}$, $\bar{n}^\mu$, $h^{\mu\nu}$ and $n_\mu$. Similar to the situation in a Newton-Cartan geometry, the geometry associated with Carrollian invariant theories is determined by the set $\bar{h}_{\mu\nu}$, $\bar{n}^\mu$ and $n_\mu$ where all $n_\mu$'s satisfying $n_\mu \bar{n}^\mu = -1$ are equivalent. This equivalence is made manifest by introducing a Carrollian version of the Milne transformation, [11], which we will refer to as a C-Milne transformation for short.[4] In analogy to the situation in Newton-Cartan geometries, a Carrollian boost is a combination of a coordinate transformation and a C-Milne transformation. Likewise, $\bar{h}_{\mu\nu}$ (and $\bar{n}^\mu$) are Carrollian covariant while $h^{\mu\nu}$ (and $n_\mu$) are not. We conclude that if we construct $\Omega^2$ from the inverse metric $h^{\mu\nu}$, it is bound to behave non covariantly under Carrollian transformations. We refer the reader to Appendix A for a concise summary of Carrollian geometry or to [11] for an extended discussion.

While $h^{\mu\nu}$ and $n_\mu$ are not Carrollian covariant, we may construct a modified inverse metric $\tilde{h}^{\mu\nu} = h^{\mu\nu} - M_C^\mu \bar{n}^\nu - M_C^\nu \bar{n}^\mu + M_C^2 \bar{n}^\mu \bar{n}^\nu$ and a modified one-form $\tilde{n}_\mu = n_\mu - \bar{h}_{\mu\nu} M_C^\nu$ which are Carrollian covariant. Here $M_C^\mu$ is an additional vector field available in geometries associated with Carrollian symmetry, similar to the gauge field $A_\mu^G$ which appears in Newton-Cartan geometries. It originates in the ambiguity in defining a symmetric, metric compatible connection. The transformation laws for this additional field, $M_C^\mu$, are given

---

[4] What we refer to as a Carrollian version of the Milne transformation was termed a local Carrollian transformations in [11]. Our construction of these transformations is somewhat different from that of [11].

in (130) and (131). Using these transformation laws it is straightforward to check that, indeed, $\tilde{h}^{\mu\nu}$ and $\tilde{n}_\mu$ are covariant tensors under Carrollian transformations. By replacing $h^{\mu\nu}$ with $\tilde{h}^{\mu\nu}$ and $n_\mu$ with $\tilde{n}_\mu$ throughout this section, (and by replacing the connection (8) with an $\tilde{h}^{\mu\nu}$ compatible one), $\Omega^2$ will be a Carrollian scalar.

Going back to (31) we find that $f$ and $g$ must be such that (31) takes the form

$$\Omega = d\left(f_C \bar{u}_C\right) + d\left(g_C(n - M_C)\right) \tag{85}$$

where $f_C = f(s_C, \tilde{n})\sqrt{v^2}$ and $\bar{u}_C = u_{C\,\mu}dx^\mu$. This is naturally satisfied by the first branch of solutions, (79), and also by the second branch of solutions, (80), once we set $f = f_0/\sqrt{v^2}$. Note that by doing so, the second branch of solutions is a special case of the first. One can now follow the same analysis as in the relativistic case to obtain $f_C = T_C f_c(T_C/\mu_C)$ and $P = P(T_C f_c(T_C/\mu_C))$ with $T_C = T/\sqrt{v^2}$ and $\mu_C = \mu/\sqrt{v^2}$.

Let us summarize. The Carrollian covariant 2-form $\Omega = \Omega_{\mu\nu}dx^\mu dx^\nu$ satisfying (14) is given by (85). If we raise its indices using the Carrollian covariant metric $\tilde{h}^{\mu\nu}$ then $\Omega^2$ is invariant under Carrollian boosts.

## 3.4 Lifshitz symmetry

Apart from enhancing the (spacetime) translation and (spatial) rotation invariant dynamics to a boost invariant one, it is also possible to add a scaling symmetry. Lifshitz symmetry is a scaling symmetry whereby the time and space coordinates are scaled differently, $t \to \lambda^z t$, $\vec{x} \to \lambda \vec{x}$. Once again, following [7], Lifshitz scale symmetry implies that

$$dP = zG + (z-1)\rho v^2 \tag{86}$$

with $d$ the number of space dimensions ($d = 2$ in this work). Equation (86) amounts to

$$G = s^{\frac{2+z}{2}} G_0\left(\tilde{n},\, v^2 s^{1-z}\right). \tag{87}$$

This implies that in a Lifshitz invariant theory the solution to (14) takes the form (41) with

$$H = -s\xi(\tilde{n}) + (s\xi(\tilde{n}))^{\frac{2+z}{2}} h\left(v^2\,(s\xi(\tilde{n}))^{1-z}\right), \qquad \eta = \xi, \tag{88}$$

or (44) with

$$J = (v^2)^{\frac{z}{2(z-1)}} j(\tilde{n}), \qquad P_0 = 0, \tag{89}$$

for $z \neq 1$. When $z = 1$ the solution (44) is trivial. (Curiously, if we set $z = -2$ then $P_0 \neq 0$ is allowed.)

One can now impose Lifshitz invariance in addition to boost invariance. Lorentz invariance is compatible only with $z = 1$ scaling leading to

$$f_L = -\frac{3}{4}\sqrt{s_L\xi(\tilde{n})}H_0, \qquad g/\gamma = f_L, \qquad G = (s_L\xi(\tilde{n}))^{3/2}H_0, \tag{90}$$

where $H_0$ is a constant and we have omitted constant terms in $g$ which do not contribute to the enstrophy current. Galilean invariant fluids are compatible only with $z = 2$ (this follows by requiring both (86) and (48) as has been pointed out in [7,31]) leading to

$$f = f_0, \qquad g = \frac{1}{2}f_0 v^2 + \frac{2f_0 H_0}{m}n, \qquad G = n^2 H_0 - \frac{1}{2}mnv^2. \tag{91}$$

Finally, equations (14) are satisfied by Carrollian invariant Lifshitz fluids if

$$f_C = \frac{1}{2}\frac{(2+z)}{(1+z)}(s_C\xi(\tilde{n}))^{\frac{1}{1+z}}H_0, \qquad g = 0, \qquad G = (s_C\xi(\tilde{n}))^{\frac{2+z}{1+z}}H_0. \tag{92}$$

A summary of these results can be found in tables 1 and 2.

# 4 Solving $i_u\Omega = 0$ for incompressible flow

Strictly speaking, all fluids are compressible. Yet, most day-to-day fluid flows, from the stream of water in a garden hose to automotive aerodynamics, are described by the incompressible Navier-Stokes equations. Indeed, under the assumption of subsonic or low Mach number flow, the Galilean invariant compressible fluid equations of motion reduce to the incompressible Navier-Stokes equations. This limiting behavior makes the latter a robust and well studied approximation of a wide variety of commonplace physical phenomena.

Of particular relevance to the current work is that incompressible Galilean fluid flow supports an enstrophy current regardless of the equation of state. As we will see shortly, the low Mach number limit of non frame invariant fluids also leads to incompressible flow which also supports an enstrophy current independent on the equation of state.

To start, let us consider the fluid equations (30) with the rescaling

$$v^i \to V_0 \hat{v}^i, \qquad t \to \frac{L_0}{V_0}\hat{t}, \qquad x \to L_0 \hat{x}, \tag{93}$$

where hatted quantities are dimensionless. Inserting (93) into (30) we find

$$0 = \hat{a}_\mu + \frac{1}{V_0^2}\frac{1}{\rho}\left(\left(\frac{\partial P}{\partial s}\right)_{n,v^2}\hat{D}_\mu^\perp s + \left(\frac{\partial P}{\partial n}\right)_{s,v^2}\hat{D}_\mu^\perp n\right) + \frac{1}{\rho}\left(\frac{\partial P}{\partial v^2}\right)_{s,n}\hat{D}_\mu^\perp \hat{v}^2 + \frac{\hat{\hat{v}}_\mu}{\rho}\left(\rho\hat{\Theta} + \hat{v}^\nu\hat{\partial}_\nu\rho\right)$$

$$0 = \hat{D}n + n\hat{\Theta} + \hat{v}^\mu\hat{D}_\mu^\perp n$$

$$0 = \hat{D}s + s\hat{\Theta} + \hat{v}^\mu\hat{D}_\mu^\perp s. \tag{94}$$

where hatted quantities are dimensionless versions of their unhatted counterparts, viz., $\hat{\Theta} = \hat{\nabla} \cdot \hat{v}$.

The speed of sound of the fluid may be computed by considering linearized perturbations of a uniform, equilibrated, configuration (see [7]). At low velocities it is given by

$$V_s^2 = \frac{1}{\rho}\left(\left(\frac{\partial P}{\partial s}\right)_s s + \left(\frac{\partial P}{\partial n}\right)_s n\right). \tag{95}$$

Expanding the equations of motion and dynamical variables, $s$, $n$ and $\hat{v}^i$ around small $M = V_0/V_s$ we find

$$0 = \left(\frac{\partial P^{(0)}}{\partial s^{(0)}}\right)_n^{(0)}\hat{D}_\mu^\perp s^{(0)} + \left(\frac{\partial P^{(0)}}{\partial n^{(0)}}\right)_s^{(0)}\hat{D}_\mu^\perp n^{(0)}$$

$$0 = \hat{D}n^{(0)} + n^{(0)}\hat{\Theta}^{(0)} + \hat{v}^{(0)\mu}\hat{D}_\mu^\perp n^{(0)} \tag{96}$$

$$0 = \hat{D}s^{(0)} + s^{(0)}\hat{\Theta}^{(0)} + \hat{v}^{(0)\mu}\hat{D}_\mu^\perp s^{(0)}$$

where we have defined,

$$s = s^{(0)} + M^2 s^{(2)} + \dots, \qquad n = n^{(0)} + M^2 n^{(2)} + \dots, \qquad \hat{v}^2 = V_0^2\left(\hat{v}^{(0)}\right)^2 + \dots, \tag{97}$$

and

$$P = P^{(0)}\left(s^{(0)}, n^{(0)}\right) + M^2\left(P^{(2)}\left(s^{(0)}, n^{(0)}; s^{(2)}, n^{(2)}\right) + V_s^2 P_{v^2}^{(2)}\left(s^{(0)}, n^{(0)}\right)\left(\hat{v}^{(0)}\right)^2\right) + \dots. \tag{98}$$

Further assuming that the particle number is constant to leading order in $M$,

$$\hat{D}n^{(0)} = 0\,, \qquad \hat{D}^{\perp}_{\mu}n^{(0)} = 0\,, \tag{99}$$

implies that

$$\hat{\Theta}^{(0)} = 0\,, \qquad \hat{D}^{\perp}_{\mu}s^{(0)} = 0\,, \qquad \hat{D}s^{(0)} = 0\,, \tag{100}$$

and therefore that $\rho^{(0)}$ and $P^{(0)}$ are constant as well. The leading order equations for the velocity field now become

$$0 = \hat{a}^{(0)}_{\mu} + \frac{1}{\rho^{(0)}}\left(\hat{D}^{\perp}_{\mu}P^{(2)}\right) + \frac{P^{(2)}_{v^2}}{\rho^{(0)}}\hat{D}^{\perp}_{\mu}\left(\hat{v}^{(0)}\right)^2\,,$$
$$0 = \hat{\Theta}^{(0)}\,. \tag{101}$$

Note that (101) is a set of 3 equations for three unknowns: $\hat{v}^{(0)i}$ and $\hat{P}^{(2)}$.

We can now go through an analysis similar to that of the previous section, in order to obtain an enstrophy current which is independent of the equation of state. That is, look for an $f(\hat{s}^{(2)}, \hat{n}^{(2)}, (\hat{v}^{(0)})^2)$ and a $g(\hat{s}^{(2)}, \hat{n}^{(2)}, (\hat{v}^{(0)})^2)$, defined in (31), which solve (14b) under the equations of motion (101). We find

$$f = f(v^2)\,, \qquad g = f(v^2)v^2 + \left(\frac{P^{(2)}_{v^2}}{\rho^{(0)}} - \frac{1}{2}\right)\int f(v^2)dv^2\,. \tag{102}$$

The result presented in (102) is inline with the known behavior of incompressible Galilean invariant fluids once we enforce $f = f_0$ a constant due to Galilean invariance. The velocity field of a relativistic fluid flowing subsonically is usually too low to exhibit relativistic effects and so it is less interesting from a physical standpoint. It is even less clear how subsonic Carrollian flow would manifest.

## 5  Summary

Our main result in this work has been to provide an operative technique to compute a putative enstrophy current in $2 + 1$ dimensional flow with varying amounts of symmetry. We used our technique to compute the enstrophy current of a non frame invariant fluid both for a generic flow, in which case the enstrophy current exists only for special equations of state, and for incompressible flow where the equation of state is unconstrained. By taking various limits of this result we managed to recover or discover how the enstrophy current behaves in Galilean, Lorentz and Carrollian invariant fluids and in fluids with an additional Lifshitz scale symmetry. Our results are summarized in tables 1 and 2.

Carrollian invariance is perhaps the least familiar limit of Lorentz invariance. Recall that the Carrollian invariant frame transformation is obtained by taking the $c \to 0$ limit of Lorentz transformations for tachyonic observers moving at superluminal velocities. Going on a slight detour, we note that it is also possible to take the Carrollian limit, where the lightcone collapses to the time axis, in such a way that velocities of massive particles vanish sufficiently fast so as to retain some of their dynamics. If we take the $c = 0$ limit of Lorentz transformations associated with observers moving at subluminal velocities and make the replacement $\vec{v} \to \vec{\nu}c^2$ then we obtain a Carrollian transformation. It would be interesting to find a dimensionless control parameter about which this limit could be expanded. In any case, in this limit we are describing the dynamics of observers trapped inside the lightcone. (Note that one can take the same type of limit to describe tachyonic observers trapped on the $t = 0$ plane when taking the Galilean, $c \to \infty$, limit.)

| Flow type | $f$ | $\bar{u}_\mu$ |
|---|---|---|
| Non frame invariant (I) | $\frac{1}{s\xi}\left(\frac{\partial G}{\partial v^2}\right)$ | $(0, v_i)$ |
| Non frame invariant (II) | $f(v^2)$ | $(0, v_i)$ |
| Galilean | $f_0$ | $(0, v_i)$ |
| Lorentzian | $-\frac{1}{2}\left(\frac{\partial G}{\partial x}\right)\Big|_{x=s\xi}$ | $\frac{1}{\sqrt{1-v^2}}(-1, v_i)$ |
| Carrollian | $\frac{1}{2}\left(\frac{\partial G}{\partial x}\right)\Big|_{x=s\xi}$ | $\frac{1}{\sqrt{v^2}}(0, v_i)$ |
| Incompressible | $f(v^2)$ | $(0, v_i)$ |

Table 1: Values of $f$ and $\bar{u}_\mu$ which determine $\Omega_{\mu\nu} = \partial_\mu(f\bar{u}_\nu) - \partial_\nu(f\bar{u}_\mu)$ from which a conserved enstrophy current can be constructed and the equation number where the value of $f$ was determined. Here, $G$ is a free energy related to the pressure, $P$, via $G + P = sT + n\mu$ with $s$ and $n$ the entropy density and charge density, and $T$ and $\mu$ the temperature and chemical potential. The explicit expression for $G$ in each of the above cases can be found in table 2. Of particular relevance to us is its dependence on $s\xi$ where $\xi$ is an arbitrary function of $\tilde{n} = n/s$.

Keeping track of factors of $c$ one finds that the velocity field associated with particles trapped inside the shrinking lightcone is given by $u^\mu = (1, 0, 0, 0) + \mathcal{O}(c^2)$ and $\bar{u}_\mu = \mathcal{O}(c^2)$. However, when taking the same $c \to 0$ limit of the stress tensor for subluminal fluid motion one finds a non trivial dependence on velocity $v_i$ due to cancellation of factors of $c^2$ in the stress tensor and in subleading components of $u^\mu$ and $\bar{u}_\mu$. The dynamics of such gasses have been described in [28]. Since the Carrollian invariant velocity field in this case is constant, $u^\mu = (1, 0, 0, 0)$, it is not clear if there is a sense in which there exists an enstrophy current even in this somewhat degenerate setting.

So far, we have only considered an approximately conserved enstrophy current. In Galilean invariant fluids the enstrophy current given by (1) has a negative definite divergence once dissipative corrections are taken into account [12]. This property, together with energy conservation, leads to an inverse energy cascade in 2+1 dimensional turbulence [1]. It is unclear at this point whether one can systematically construct an Aristotelian enstrophy current with a sign-definite divergence. If such a construction exists, it will shed light on the role of enstrophy in 2+1 dimensional turbulent flow with varying symmetry.

The existence of a relativistic enstrophy current in $2 + 1$ dimensional flow, implies the existence of a dual quantity in a holographic description of fluid flow in $3 + 1$ dimensional gravity. Like entropy, the gravitational manifestation of enstrophy may persist beyond asymptotically AdS black brane geometries. We leave such issues for future work.

## Acknowledgments

We would like to thank A. Frishman for useful discussions and W. Sybesma for discussions and comments on a draft of this manuscript. NPF is supported by the European Commission through the Marie Sklodowska-Curie Action UniCHydro (grant agreement ID: 886540). AY is supported in part by an Israeli Science Foundation excellence center grant 2289/18 and a Binational Science Foundation grant 2016324.

| Flow type | $G$ | $u^\mu$ | $g^{\mu\nu}$ |
|---|---|---|---|
| Non frame invariant (I) | $H(s\xi(\tilde{n}),\, v^2) + s\eta(\tilde{n})$ | $(1, v^i)$ | $h^{\mu\nu}$ |
| Non frame invariant (II) | $-P_0 + sJ(\tilde{n},\, v^2)^\dagger$ | $(1, v^i)$ | $h^{\mu\nu}$ |
| Galilean | $H_G(s\tilde{n}) + s\eta(\tilde{n}) - \frac{1}{2}ms\tilde{n}v^2$ | $(1, v^i)$ | $h^{\mu\nu}$ |
| Lorentzian | $H_L(s\xi(\tilde{n}))$ | $\frac{(1, v^i)}{\sqrt{1-v^2}}$ | $h^{\mu\nu} - \bar{n}^\mu\bar{n}^\nu$ |
| Carrollian | $H_C(s\xi(\tilde{n}))$ | $\frac{(1, v^i)}{\sqrt{v^2}}$ | $h^{\mu\nu\ddagger}$ |
| Non frame invariant Lifshitz (I) | $(s\xi(\tilde{n}))^{\frac{2+z}{2}}\, h\left(v^2\, (s\xi(\tilde{n}))^{1-z}\right)$ | $(1, v^i)$ | $h^{\mu\nu}$ |
| Non frame invariant Lifshitz (II) $(z \neq 1)$ | $-P_0 + s\left(v^2\right)^{\frac{z}{2(z-1)}} j(\tilde{n})$ | $(1, v^i)$ | $h^{\mu\nu}$ |
| Galilean Lifshitz $(z = 2)$ | $n^2 H_0 - \frac{1}{2}mnv^2$ | $(1, v^i)$ | $h^{\mu\nu}$ |
| Lorentzian Lifshitz $(z = 1)$ | $(s\xi(\tilde{n}))^{\frac{3}{2}} H_0$ | $\frac{(1, v^i)}{\sqrt{1-v^2}}$ | $h^{\mu\nu} - \bar{n}^\mu\bar{n}^\nu$ |
| Carrollian Lifshitz | $(s\xi(\tilde{n}))^{\frac{2+z}{1+z}} H_0$ | $\frac{(1, v^i)}{\sqrt{v^2}}$ | $h^{\mu\nu\ddagger}$ |

Table 2: Constraints on the form of the free energy $G$, the velocity field, $u^\mu$ and $\Omega^2 = \Omega_{\mu\nu}\Omega_{\rho\sigma}g^{\mu\rho}g^{\nu\sigma}$ needed in order to construct a conserved enstrophy current $J^\mu = q\left(\frac{s}{n}\right)\frac{(\Omega^2)^\alpha}{s^{2\alpha-1}}u^\mu$. Here, $G$ is a free energy related to the pressure, $P$, via $G + P = sT + n\mu$ with $s$ and $n$ the entropy density and charge density, and $T$ and $\mu$ the temperature and chemical potential. We have also used the shorthand $\tilde{n} = n/s$. We have not included an entry for incompressible flow where the equation of state is arbitrary and the velocity field and inverse metric take on their non frame invariant, Lorentz invariant, Galilean invariant, or Carrollian invariant form.

$\dagger$ $J$ must satisfy $\partial_{v^2}G + 2v^2\partial_{v^2}G \neq 0$.

$\ddagger$ This is the expression for the inverse metric after setting $M^C = 0$. Otherwise the inverse metric is given by (7d).

# A   Boost invariance

In this section we briefly review some salient features of Galilean and Carrollian boost invariance paraphrasing the results of [14, 25] and [11].

## A.1   Galilean boosts

The Lorentz group is, by definition, represented by those coordinate transformations under which the Minkowski metric is invariant. In a similar vein, a representation of the Galilean group may be defined as those transformations under which the flat spatial metric and its accompanying geometric data remain invariant. Recall that an Aristotelian geometry is given by a degenerate inverse metric, $h^{\mu\nu}$, satisfying $h^{\mu\nu}n_\nu = 0$, and a preferred time direction $\bar{n}^\mu$ normalized such that $\bar{n}^\mu n_\mu = -1$. From these one may construct the metric $\bar{h}_{\mu\nu}$ as discussed in section 2. A flat Aristotelian geometry, in Cartesian coordinates, is given by (5).

A Newton-Cartan geometry is also equipped with a degenerate inverse metric $h^{\mu\nu}$ and its associated eigenvector $n_\mu$, but instead of a time direction one considers an equivalence class of time directions $\bar{n}^\mu$ where $\bar{n}_1^\mu \sim \bar{n}_2^\mu$ as long as $\bar{n}_1^\mu n_\mu = \bar{n}_2^\mu n_\mu = -1$. In addition, Newton-Cartan geometry possesses additional data without which a unique, symmetric, metric compatible and Galilean covariant connection can not be specified. The minimal requirements for defining such a connection is to introduce a covector $M_\mu^G$ which transforms appropriately under Galilean boosts. Often, $M_\mu^G$ is identified with the gauge field $A_\mu^G$

associated with particle number conservation. In what follows we will elaborate on the equivalence class associated with time directions mentioned above and on the roles played by $M_\mu^G$ and $A_\mu^G$.

To make the equivalence $\bar{n}_1^\mu \sim \bar{n}_2^\mu$ manifest we allow the geometric data to transform under a Milne transformation parameterized by a covector $\psi_\nu$,

$$\bar{n}^\mu \xrightarrow[G-Milne]{} \bar{n}^\mu + h^{\mu\nu}\psi_\nu\,. \tag{103a}$$

The prefactor 'G' is a reminder that we are referring to the Galilean version of the Milne transformation, distinct from its Carrollian version to be discussed in the next section. The transformation (103a) implies

$$\bar{h}_{\mu\nu} \xrightarrow[G-Milne]{} \bar{h}_{\mu\nu} + n_\mu P_\nu^\alpha \psi_\alpha + n_\nu P_\mu^\alpha \psi_\alpha + \psi^2 n_\mu n_\nu\,, \tag{103b}$$

with $\psi^2 = h^{\mu\nu}\psi_\mu\psi_\nu$, and $P_\nu^\mu = \delta_\nu^\mu + \bar{n}^\mu n_\nu$. The inverse metric $h^{\mu\nu}$ and its eigenvector $n_\mu$ are taken to be inert under Milne transformations.

As mentioned earlier, the Galilean group can be represented by those coordinate transformations $x^\mu \to x'^\mu(x)$ and Milne transformations with parameter $\psi_\nu$ which keep the flat Cartesian Newton-Cartan geometric data (5) invariant. The Galilean boosts are a subset of these transformations satisfying

$$x'^\mu = (t, \vec{x} - \vec{v}_0\,t)\,, \qquad \psi_\mu = (0, \vec{v}_0)\,, \tag{104}$$

with constant $\vec{v}_0$. In general coordinates, Galilean boosts are given by

$$G^\mu{}_\nu \equiv \partial x'^\mu/\partial x^\nu = \delta_\nu^\mu + h^{\mu\alpha}\lambda_\alpha n_\nu\,, \qquad \psi_\mu = \lambda_\mu\,, \tag{105}$$

where $\lambda_\mu$ is spacetime dependent and reduces to $\lambda_\mu = (0, \vec{v}_0)$ in a Cartesian coordinate system.

Coordinate transformations associated with $G^\mu{}_\nu$ defined in (105) act on tensors in the standard way,

$$\begin{aligned} T^{\mu_1\ldots\mu_p}{}_{\nu_1\ldots\nu_q} \xrightarrow[Coordinate]{} & T^{\mu_1\ldots\mu_p}{}_{\nu_1\ldots\nu_q} + \bar{\lambda}^{\mu_1} n_\alpha T^{\alpha\ldots\mu_p}{}_{\nu_1\ldots\nu_q} + \cdots + \bar{\lambda}^{\mu_p} n_\alpha T^{\mu_1\ldots\alpha}{}_{\nu_1\ldots\nu_q} \\ & - n_{\nu_1}\bar{\lambda}^\alpha T^{\mu_1\ldots\mu_p}{}_{\alpha\ldots\nu_q} - \cdots - n_{\nu_q}\bar{\lambda}^\alpha T^{\mu_1\ldots\mu_p}{}_{\nu_1\ldots\alpha} + \cdots \end{aligned} \tag{106}$$

where we have defined $\bar{\lambda}^\mu = h^{\mu\alpha}\lambda_\alpha$ and the last $\ldots$ denote nonlinear terms in $\lambda$, e.g.,

$$\begin{aligned} T^{\mu\nu} &\xrightarrow[Coordinate]{} T^{\mu\nu} + \bar{\lambda}^\mu n_\alpha T^{\alpha\nu} + \bar{\lambda}^\nu n_\alpha T^{\mu\alpha} + \bar{\lambda}^\mu \bar{\lambda}^\nu (T^{\alpha\beta} n_\alpha n_\beta) \\ O_{\mu\nu} &\xrightarrow[Coordinate]{} O_{\mu\nu} - n_\mu \bar{\lambda}^\alpha O_{\alpha\nu} - n_\nu \bar{\lambda}^\alpha O_{\mu\alpha} + n_\mu n_\nu (O_{\alpha\beta}\bar{\lambda}^\alpha \bar{\lambda}^\beta)\,. \end{aligned} \tag{107}$$

We define a tensor to be Galilean covariant if it transforms as in (106) under Galilean boosts, (105). Thus, $n_\mu$ and $h^{\mu\nu}$ are Galilean covariant (and also invariant under Galilean transformations). Contrarily, $\bar{n}^\mu$ and $\bar{h}^{\mu\nu}$ do not transform covariantly under Galilean boosts. Instead we find

$$\bar{n}^\mu \xrightarrow[Galilean]{} \bar{n}^\mu\,, \qquad \bar{h}_{\mu\nu} \xrightarrow[Galilean]{} \bar{h}_{\mu\nu} + 2\lambda^2 n_\mu n_\nu\,, \tag{108}$$

where $\lambda^2 = \lambda_\alpha\lambda_\beta h^{\alpha\beta}$.

As opposed to a Lorentzian geometry, in Newton-Cartan geometry, metric compatibility (with respect to $h^{\mu\nu}$ and $n_\mu$) does not uniquely specify the symmetric part of the

connection $\Gamma^{\rho}_{\mu\nu}$. In the torsionless case, a computation similar to the one carried out in [14] leads to

$$\Gamma^{\mu}_{G\nu\rho} = -\bar{n}^{\mu}\partial_{\rho}n_{\nu} + \frac{1}{2}h^{\mu\sigma}\left(\partial_{\nu}\bar{h}_{\rho\sigma} + \partial_{\rho}\bar{h}_{\nu\sigma} - \partial_{\sigma}\bar{h}_{\nu\rho}\right) + h^{\mu\sigma}n_{(\nu}F_{\rho)\sigma} \tag{109}$$

with $F^G = dM_{(G)}$. In order to ensure that the covariant derivative associated with (109) is compatible with Galilean invariance we must associate to $M^G_{\mu}$ a Milne transformation of the form

$$M^G_{\mu} \xrightarrow[G-Milne]{} M^G_{\mu} + P^{\alpha}_{\mu}\psi_{\alpha} + \frac{1}{2}n_{\mu}\psi^2. \tag{110}$$

We note in passing that Galilean covariance of the new connection (109) can be made manifest by rewriting it in the form

$$\Gamma^{\mu}_{G\nu\rho} = -\tilde{n}^{\mu}\partial_{\rho}n_{\nu} + \frac{1}{2}h^{\mu\sigma}\left(\partial_{\nu}\tilde{h}_{\rho\sigma} + \partial_{\rho}\tilde{h}_{\nu\sigma} - \partial_{\sigma}\tilde{h}_{\nu\rho}\right), \tag{111}$$

where

$$\tilde{n}^{\mu} = \bar{n}^{\mu} - h^{\mu\sigma}M^G_{\sigma}, \qquad \tilde{h}_{\mu\nu} = \bar{h}_{\mu\nu} - n_{\mu}P^{\alpha}_{\nu}M^G_{\alpha} - n_{\nu}P^{\alpha}_{\mu}M^G_{\alpha} + (M^G)^2 n_{\mu}n_{\nu}, \tag{112}$$

and $(M^G)^2 = M^G_{\mu}M^G_{\nu}h^{\mu\nu}$. It is straightforward to check that $\tilde{n}^{\mu}$ and $\tilde{h}_{\mu\nu}$ are Milne invariant and therefore also Galilean covariant. For completeness we note that one may use (112) to define the connection

$$\tilde{\Gamma}^{\mu}_{G\nu\rho} = -\tilde{n}^{\mu}\partial_{\rho}n_{\nu} + \frac{1}{2}h^{\mu\sigma}\left(\partial_{\nu}\tilde{h}_{\rho\sigma} + \partial_{\rho}\tilde{h}_{\nu\sigma} - \partial_{\sigma}\tilde{h}_{\nu\rho}\right) + \frac{1}{2}h^{\mu\sigma}n_{\rho}\pounds_{\tilde{n}}\tilde{h}_{\sigma\nu}, \tag{113}$$

which generates a Galilean covariant connection compatible with $n_{\mu}$, $h^{\mu\nu}$, $\tilde{n}^{\mu}$ and $\tilde{h}_{\mu\nu}$.

Often one considers the Galilean group associated with the massive Galilean, or Bargmann, algebra. In that case the background geometry includes an additional gauge field $A^G_{\mu}$ associated with the $U(1)$ symmetry responsible for particle number, or mass, conservation. This gauge field transforms as a covector under coordinate transformations associated with Galilean boosts,

$$A^G_{\mu} \xrightarrow{Coordinate} A^G_{\mu} - n_{\mu}\left(A^G \cdot \bar{\lambda}\right), \tag{114}$$

as a connection under $U(1)$ gauge transformations,

$$A^G_{\mu} \xrightarrow[Gauge]{} A^G_{\mu} - \partial_{\mu}\Lambda, \tag{115}$$

and if we use $A^G_{\mu}$ in place of $M^G_{\mu}$ to specify the connection, as is often done in the literature, then it transforms inhomogenously under Milne transformations,

$$A^G_{\mu} \xrightarrow[G-Milne]{} A^G_{\mu} + P^{\alpha}_{\mu}\psi_{\alpha} + \frac{1}{2}n_{\mu}\psi^2. \tag{116}$$

As observed in [14], $A^G_{\mu} = 0$ is invariant under the combination of a Galilean boost (105) and a gauge transformation (115) with parameter

$$\Lambda = \int\left(\lambda_{\mu} + \frac{1}{2}\lambda^2 n_{\mu}\right)dx^{\mu}, \tag{117}$$

where

$$\partial_{\mu}\partial_{\nu}\Lambda - \partial_{\nu}\partial_{\mu}\Lambda = 0. \tag{118}$$

In a torsionless background, the latter condition is satisfied if $\lambda_{\mu}$ and $n_{\mu}$ are covariantly constant.

Of particular importance to this work is the Galilean velocity field $u_G^\mu = \partial x^\mu / \partial t$ which transforms as a vector under coordinate transformations associated with Galilean boosts,

$$u_G^\mu \xrightarrow[Coordinate]{} u_G^\mu + \bar\lambda^\mu (u_G \cdot n) = u_G^\mu - \bar\lambda^\mu \tag{119}$$

and is inert under Milne transformations,

$$u_G^\mu \xrightarrow[G-Milne]{} u_G^\mu \,. \tag{120}$$

While $u_G^\mu$ transforms as a vector under Galilean boosts (105), $\bar u_{G\,\mu} = \bar h_{\mu\nu} u_G^\nu$ and $u_G^2$ do not,

$$\begin{aligned}
\bar u_{G\,\mu} &\xrightarrow[Galilean]{} \bar u_{G\,\mu} - \lambda_\mu - n_\mu \lambda^2 \,, \\
u_G^2 &\xrightarrow[Galilean]{} u_G^2 + \lambda^2 - 2(\bar u \cdot \bar\lambda) \,.
\end{aligned} \tag{121}$$

The Galilean covariance of $\bar u_{G\,\mu}$ is spoiled by the nontrivial Milne transformation properties of $\bar h_{\mu\nu}$ given in (103). While $\bar u_{G\,\mu}$ is not a Galilean covariant vector, it is straightforward to check that gauge invariant expressions constructed out of the combination $A_\mu^G + \bar u_{G\,\mu} + \frac{1}{2} n_\mu u_G^2$ are Galilean covariant (see, e.g., [14]). This is the reason that in (57) we replaced $d(f^G \bar u_G)$ with the Galilean covariant expression $f_0 d\left(A_\mu^G + \bar u_{G\,\mu} + \frac{1}{2} n_\mu u_G^2\right)$ with $f_0$ a constant.

## A.2 Carrollian boosts

The Carrollian equivalent of Newton-Cartan geometry includes a degenerate metric $\bar h_{\mu\nu}$ satisfying $\bar h_{\mu\nu} \bar n^\mu = 0$, an equivalence class of normals $n_\mu^1 \sim n_\mu^2$, and an extra field $M_C^\mu$ associated with the Carrollian connection. As was the case in Newton-Cartan geometry, the equivalence class between normals may be made manifest by introducing a Carrollian version of the Milne transformation which leaves $\bar h_{\mu\nu}$ and $\bar n^\mu$ invariant and transforms $n_\mu$ as

$$n_\mu \xrightarrow[C-Milne]{} n_\mu + \bar h_{\mu\nu} \phi^\nu \,, \tag{122a}$$

implying

$$h^{\mu\nu} \xrightarrow[C-Milne]{} h^{\mu\nu} + \bar n^\mu P_\alpha^\nu \phi^\alpha + \bar n^\nu P_\alpha^\mu \phi^\alpha + \bar n^\mu \bar n^\nu \phi^2 \,, \tag{122b}$$

where $\phi$ is a generic spacetime dependent parameter and $\phi^2 = \phi^\mu \phi^\nu \bar h_{\mu\nu}$. The prefactor 'C' in (122) and throughout this section is used to distinguish the Carrollian version of the Milne transformation from its Galilean counterpart.

The Carrollian group is represented by coordinate transformations $x \to x'(x)$ and Carrollian-Milne (C-Milne for short) transformations with parameter $\phi^\mu$ which keep the flat Carrollian geometry invariant. In general coordinates, Carrollian boosts, which are a subset Carrollian transformations, are given by

$$C^\mu{}_\nu \equiv \frac{\partial x'^\mu}{\partial x^\nu} = \delta_\nu^\mu - \bar n^\mu \bar h_{\nu\alpha} \beta^\alpha \,, \qquad \phi^\mu = \beta^\mu \,, \tag{123}$$

where $\beta^\mu$ is a covariantly constant parameter. In flat Cartesian coordinates, where

$$\bar h_{\mu\nu} = \delta_\mu^i \delta_\nu^j \delta_{ij} \,, \qquad n_\mu = -\delta_\mu^0 \,, \tag{124}$$

the parameter $\beta^\mu$ reduces to a constant, $\beta^\mu = \frac{1}{v_0^2}(0, \vec v_0)$, and (123) reduces to

$$x^\mu = (t, \vec x) \xrightarrow[Coordinate]{} \left(t - \frac{\vec v_0 \cdot \vec x}{v_0^2}, \vec x\right) \,, \qquad \phi^\mu = \frac{1}{v_0^2}(0, \vec v_0) \,. \tag{125}$$

We define a tensor to be Carrollian covariant if

$$T^{\mu_1...\mu_p}{}_{\nu_1...\nu_q} \xrightarrow[Carrollian]{} T^{\mu_1...\mu_p}{}_{\nu_1...\nu_q} - \bar{n}^{\mu_1}\bar{\beta}_\alpha T^{\alpha...\mu_p}{}_{\nu_1...\nu_q} - \cdots - \bar{n}^{\mu_p}\bar{\beta}_\alpha T^{\mu_1...\alpha}{}_{\nu_1...\nu_q}$$
$$+ \bar{\beta}_{\nu_1}\bar{n}^\alpha T^{\mu_1...\mu_p}{}_{\alpha...\nu_q} + \cdots + \bar{\beta}_{\nu_q}\bar{n}^\alpha T^{\mu_1...\mu_p}{}_{\nu_1...\alpha} + \cdots \tag{126}$$

where we have defined $\bar{\beta}_\mu = \bar{h}_{\mu\nu}\beta^\nu$ and the last ellipses denote terms which are quadratic in $\beta^\mu$. Similar to the Galilean case, $\bar{h}_{\mu\nu}$ and $\bar{n}^\mu$ are Carrollian covariant but $n_\mu$ and $h^{\mu\nu}$ are not

$$n_\mu \xrightarrow[Carrollian]{} n_\mu\,, \qquad h^{\mu\nu} \xrightarrow[Carrollian]{} h^{\mu\nu} + 2\beta^2\bar{n}^\mu\bar{n}^\nu\,, \tag{127}$$

where $\beta^2 = \bar{h}_{\mu\nu}\beta^\mu\beta^\nu$.

To obtain a Carrollian covariant connection one can go through a construction analogous to the one which lead to (111). We will not go through it in detail but merely quote the end result. The connection

$$\Gamma^\mu_{C\nu\rho} = -\bar{n}^\mu\partial_\rho\tilde{n}_\nu + \frac{1}{2}\tilde{h}^{\mu\sigma}\left(\partial_\nu\bar{h}_{\rho\sigma} + \partial_\rho\bar{h}_{\nu\sigma} - \partial_\sigma\bar{h}_{\nu\rho}\right)\,, \tag{128}$$

with

$$\tilde{h}^{\mu\nu} = h^{\mu\nu} - P^\mu_\alpha M^\alpha_C\bar{n}^\nu - P^\nu_\alpha M^\alpha_C\bar{n}^\mu + \bar{n}^\mu\bar{n}^\nu M^2_C\,, \qquad \tilde{n}_\mu = n_\mu - \bar{h}_{\mu\alpha}M^\alpha_C \tag{129}$$

is Carrollian invariant given that

$$M^\mu_C \xrightarrow[Coordinate]{} M^\mu_C - \bar{n}^\mu(M_C \cdot \beta)\,. \tag{130}$$

and

$$M^\mu_C \xrightarrow[C-Milne]{} M^\mu_C + P^\mu_\alpha\phi^\alpha + \frac{1}{2}\bar{n}^\mu\phi^2\,. \tag{131}$$

Similar to the Galilean case, the Carrollian geometry also admits a tilde'd connection

$$\tilde{\Gamma}^\mu_{C\nu\rho} = -\bar{n}^\mu\partial_\rho\tilde{n}_\nu + \frac{1}{2}\tilde{h}^{\mu\sigma}\left(\partial_\nu\bar{h}_{\rho\sigma} + \partial_\rho\bar{h}_{\nu\sigma} - \partial_\sigma\bar{h}_{\nu\rho}\right) + \frac{1}{2}\tilde{h}^{\mu\sigma}\tilde{n}_\rho\pounds_{\bar{n}}\bar{h}_{\sigma\nu}\,, \tag{132}$$

which is compatible with $\tilde{h}^{\mu\nu}$, $h_{\mu\nu}$, $\bar{n}^\mu$ and $n_\mu$.

The Carrollian velocity field $u^\mu_C$ is Carrollian covariant. It transforms covariantly under Carrollian coordinate transformations (123),

$$u^\mu_C \xrightarrow[Coordinate]{} u^\mu_C - \bar{n}^\mu(u_C \cdot \beta)\,, \tag{133}$$

and is invariant under C-Milne transformations

$$u^\mu_C \xrightarrow[C-Milne]{} u^\mu_C\,. \tag{134}$$

Since $\bar{h}_{\mu\nu}$ transforms covariantly under Carrollian boosts, so does $\bar{u}_{C\,\mu} = \bar{h}_{\mu\nu}u^\nu_C$. The covector $n_\mu$ is not Carrollian covariant but the C-Milne invariant combination $\tilde{n}_\mu$ defined in (129) is. This justifies the use of $d(f_C\bar{u}_C) + d(g_C\tilde{n})$ in the definition of a Carrollian covariant $\Omega_{\mu\nu}$ in (85).

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
