# Peer review of "Enstrophy without boost symmetry"

_SciPost Physics_

## Round 1 · Referee Report · Anonymous · 2022-1-12

Strengths

1- concrete problem
2-complete result

Weaknesses

1-sometimes opaque due to technicalities
2-absence of toy model

Report

For a non-relativistic (Bargmann) inviscous incompressible fluid the enstrophy current is a conserved current (in two spatial dimensions) that becomes almost conserved (with a negative semi-definite sign) once viscous terms are taken into account. Enstrophy characterizes features of turbulence.

The aim of the manuscript is to drop the assumption of boost symmetry and consider only translational and spatial rotations as a symmetry of the theory (aka Aristotelian symmetries) and togerther with assuming a sensible thermodynamics limit show that conserved enstrophy currents exist and constrain their functional form.

As checks they impose Bargmann as well as Lorentz boosts on their final Aristotelian result, given in full generality, as well as considering new cases of (combined with) Lifshitz scaling and Carroll boost.

The authors provide a clear overview of the implications for the need of non-Lorentzian geometry. It is this geometry that makes this paper appear technical, but that is an inherent attribute of the topic.

I recommend this manuscript for publication.

Requested changes

I have some additional questions/comments:

- Your tables are helpful as you already refer to them in the introduction. You might want to consider connecting type (I) and (II) to the equations where they arise in the text, (37) and (44) respectively for people who first start to have a look at the tables.

- Have the authors considered to recap the canonical relativistic conformal case as e.g. 1210.6702? This computation immediately gives the uninitiated reader a clear intuition of what to do in the Aristotelian case.

- In the summary you suggest using holography to provide a 'microscopic' model. Do you expect type (I) or type (II) to be applicable?

- Can you place the microscopic toy model used e.g. in reference [7] in type (I) or (II)? Would using this toy model add something; i.e. does a more explicit expression for Enstrophy teach something?

- In the old works on the membrane paradigm by T. Damour, it was found that one can describe the black hole by pretending that the horizon is a membrane whose dynamics are captured by a fluid with Galilean symmetries. More recently it was claimed by L. Donnay and C. Marteau that this membrane actually has the dynamics of a Carrollian fluid. Do you expect Enstrophy to play a role in characterizing the black hole horizon dynamics?

- In 3+1 dimensional hydrodynamics there is a transverse plane to the direction of the magnetic field. Do you expect your analysis to be extendable to e.g. the generalized global symmetry setting of 1610.07392?

  • validity: high
  • significance: ok
  • originality: good
  • clarity: high
  • formatting: excellent
  • grammar: perfect

Author:  Natalia Pinzani-Fokeeva  on 2022-03-18  [id 2303]

(in reply to Report 1 on 2022-01-12)

We thank the referee for carefully reading our manuscript and for providing positive feedback. Here we reply to their comments:

  1. We have added more explanations to the captions of Tables 1 and 2 to address the referee's concern.

  2. We have expanded the penultimate paragraph in the introduction by adding the simple example of conformal fluids. We hope this addition satisfies the referee's concerns.

  3. Holography is best understood in the case of relativistic theories and therefore for relativistic fluids. More general gravitational backgrounds with a putative holographic dual have been constructed in the case of Lifshitz symmetry. Also, Carrollian symmetry might be relevant for the construction of holographic theories in asymptotically flat spacetimes. However, little is known for holographic duals of theories with Aristotelian symmetry (or Galilean symmetry). For this reason, we are unable to argue pro or against whether Type I and II equations of state are expected in a holographic theory.

  4. We have not analyzed whether the microscopic models of reference [7] reproduce type (I) or type (II) equations of state. We thank the referee for pointing this to us and we will consider a comparison in our future work.

  5. We do expect that enstrophy plays a role in characterizing the horizon dynamics of 3+1 dimensional black holes regardless of whether the horizon Einstein's equations are interpreted in terms of non-relativistic Navier Stokes equations a la Damour or in terms of Carrollian equations as in Donnay et al. We have in fact initiated a study of enstrophy and black hole horizons in the context of holography in 2111.00544.

  6. If the transverse plane described in 1610.07392 has a dynamics that is independent of the additional direction and can be described from some effective 2+1 dimensional fluid point of view, it might be possible to extend our results to this case. We thank the referee for this interesting food for thought.

---

## Round 1 · Referee Report · Anonymous · 2022-2-13

Strengths

1- Enstrophy is an important quantity in fluid dynmaics that is particularly useful in the study of turbulent flows. The authors provide a general prescription to construct enstrophy currents in physical systems that do not possess Galilean or Lorentzian boost symmetry.

2- The analysis in the paper, while technical, is rigorous and complete. The authors do a great job of summarising their important physical results in the introduction that softens the blow of technicality.

3- The authors spend a good amount of time analyzing various limits of their general results, which is helpful from both practical and pedagogical points of view.

4- The paper is well organized and the calculational details are presented in a systematic and understandable manner.

Weaknesses

1- The obvious weakness of the paper is its technical presentation.

2- The analysis presented by the authors is valuable as an "umbrella construction" that covers enstrophy currents for fluids with various kinds of boost symmetries. However, a discussion about the physical relevance of not having a boost symmetry is missing.

3- The covariant notation employed by the authors is neat. However, for a reader uninitiated in the ways of Aristotelian and Newton-Cartan spacetimes, it might be challenging to appreciate the results presented by the authors.

Report

I think that the paper is suitable for publication in SciPost Physics. However, I would like to suggest some minor changes to the presentation of the results to increase the readability and usefulness.

Requested changes

1- The authors should consider pulling the tables from the summary section to the introduction, so as to make the introduction self-contained for the readers who would otherwise get lost in the technicalities of the forthcoming discussion.

2- It will probably be beneficial to avoid (or minimize) the use of covariant index notation in the introduction. For instance, in eq. (4) it is not immediately clear how $\Omega^2$ is supposed to be constructed from a covariant closed 2-form $\Omega_{\mu\nu}$ in the absence of a non-degenerate metric.

3- Currently, the only statement in the paper about systems without boost symmetry is a comment on flocking in the second paragraph in the introduction. The authors should expand upon this and comment on the physical relevance and need of considering a general fluid that does not have a boost symmetry.

4- Drawing attention to any distinctive physical features of the enstropy current when no boost symmetry is present will also go a long way in asserting the physical relevance of the results.

  • validity: high
  • significance: ok
  • originality: good
  • clarity: good
  • formatting: good
  • grammar: perfect

Author:  Natalia Pinzani-Fokeeva  on 2022-03-18  [id 2302]

(in reply to Report 2 on 2022-02-13)

We thank the referee for carefully reading our manuscript and for providing positive feedback. Here we reply to their comments:

  1. We refer to Tables 2 and 3 at the end of Page 2. We prefer not to pull the tables to the introduction as we think that operation would make the presentation heavier than easier. In fact, we would need to explain all the notation and the details contained in those tables which we find unnecessary for the level of the presentation contained in the introduction. We nevertheless expanded the penultimate paragraph in the introduction with one simple example of a conserved enstrophy current, that of Lorentzian fluids. We hope this addition satisfies the referee's concerns.

  2. We have added a comment towards the end of Page 2.

  3. and 4. Our main interest in understanding the enstrophy current within the context of Aristotelian fluids was to see how robust it is. We would, of course, be extremely interested in applications, whether to flocking or other physical behavior. Unfortunately, we have not found such an application. We modified the text according to this philosophy.

---

## Editorial Decision

resubmitted